# Methods Established for *EPSPS* Gene Mutation Detection in Glyphosate-Resistant Rice (*Oryza sativa* L.)

**DOI:** 10.3390/plants14152256

**Published:** 2025-07-22

**Authors:** Xiuping Chen, Huilin Yu, Chunmeng Huang, Chenhui Hou, Haoyuan Guan, Jiajian Xie

**Affiliations:** State Key Laboratory for Biology of Plant Diseases and Insect Pests, Institute of Plant Protection, Chinese Academy of Agricultural Sciences, Beijing 100193, China; xpchen@ippcaas.cn (X.C.); hlyu@ippcaas.cn (H.Y.); cmhuang0812@163.com (C.H.); hou13213938029@163.com (C.H.); 15203802516@163.com (H.G.)

**Keywords:** glyphosate-resistant rice, *EPSPS* mutation, detection method, sensitivity

## Abstract

“Rundao118” is a glyphosate-resistant rice; it contains both endogenous wild and mutated 5-enolpyruvylshikimate-3-phosphate synthase (*EPSPS)* genes. Conventional qualitative and quantitative detection methods face significant challenges for direct analysis. Here, we describe five detection methods for identifying *EPSPS* mutations in this rice line: (1) polymerase chain reaction (PCR) amplification-based Sanger sequencing, (2) next-generation sequencing (NGS) based on PCR amplification, (3) allele-specific PCR (AS-PCR), (4) real-time fluorescent quantitative PCR (qPCR), and (5) blocker displacement amplification (BDA). All five methods effectively identified *EPSPS* mutations, with the following detection sensitivities: Sanger, 10%; NGS, 1%; AS-PCR, 0.05%; qPCR, 0.01%; and BDA, 0.1%. Among these, the Sanger, NGS, and BDA methods excelled at the rapid identification of single-nucleotide mutations, making them suitable for precise mutation site characterization and identification. In contrast, the AS-PCR and qPCR methods were more appropriate for large-scale rapid screening of known mutation sites. The detection systems established in this study provide a comprehensive technical solution for rapid identification of *EPSPS* mutations in glyphosate-resistant rice. These methods not only enable accurate determination of mutation sequences but also effectively trace mutation origins, offering crucial technical support for both safety regulations and intellectual property protection.

## 1. Introduction

Glyphosate (C_3_H_8_NO_5_P) is a highly effective broad-spectrum, non-selective, and low-residue herbicide that is widely used in agricultural production for weed control [1]. Its primary mechanism of action involves inhibiting the enzyme in plants, thereby exerting its herbicidal effect [2,3]. The rice *EPSPS* gene is sensitive to glyphosate, making conventional rice non-tolerant to the herbicide. Mutations at amino acid positions 172–174 and 177 (G172A, T173I, A174V, P177S) in rice EPSPS are key to conferring glyphosate tolerance in rice, thus targeted mutations at key sites in EPSPS can confer glyphosate tolerance in rice [4,5,6,7,8].

Two strategies are generally used for developing herbicide-resistant rice through *EPSPS* modification: the first involves in vivo gene editing of *OsEPSPS*. For example, Gao et al. [9,10] and Xia et al. [11] used CRISPR-Cas9 gene-editing technologies to mutate endogenous *EPSPS* in rice, successfully generating glyphosate-resistant rice. The second strategy entails in vitro mutagenesis of *EPSPS* followed by its introduction into rice. For instance, Sichuan Tianyu Xinghe Biotechnology Co., Ltd. (Chengdu, China) used “Gene Super Evolution” technology to modify rice *EPSPS* in vitro and then introduced the engineered gene into conventional rice varieties to develop a series of glyphosate-tolerant rice lines [12]. A sequence comparison of key nucleotide mutations in *EPSPS* genes between wild-type and modified variants is shown in Figure 1.

Depending on the research requirements, such as the mutation type, detection sensitivity, throughput, and cost, the following methods can be flexibly used for detecting genetic mutations: (1) sequencing-based methods, such as Sanger sequencing and high-throughput sequencing, e.g., NGS and third-generation sequencing [13]; (2) PCR-based methods, including qualitative PCR, e.g., AS-PCR, restriction fragment length polymorphism, and quantitative PCR, e.g., qPCR and droplet digital PCR (ddPCR); and (3) hybrid sequencing–PCR methods, such as amplicon sequencing and BDA [14].

Sanger sequencing is the traditional gold standard for gene mutation detection [15] and is suitable for accurate analysis of known mutations, though with limited throughput. In contrast, NGS, a second-generation sequencing technology, enables high-throughput and rapid and efficient large-scale parallel DNA sequencing. As compared with first-generation sequencing (e.g., Sanger sequencing), NGS offers notable advantages in speed, throughput, and cost-effectiveness, making it widely applicable in genomics, transcriptomics, epigenetics, and related fields [13,16].

Qualitative PCR, such as AS-PCR, can be used for rapid identification of the presence or absence of specific mutations. qPCR is currently one of the most widely used quantitative detection technologies and is primarily divided into probe-based and dye-based methods. This technique involves adding fluorescent dyes or sequence-specific fluorescent probes to the reaction system. As PCR amplification progresses, the accumulation of products leads to an increase in fluorescence, allowing real-time monitoring of the reaction. During the exponential phase of PCR amplification, the cycle threshold (Ct) value exhibits a linear relationship with the logarithm of the initial template copy number, which serves as the basis for quantification [17,18].

BDA is an allele-specific PCR enrichment technology that leverages thermodynamic differences in the binding of primers, blocking probes, and template DNA during PCR amplification. This approach suppresses the amplification of wild-type templates while enhancing signals from low-frequency mutations, thereby improving detection sensitivity for rare variants [19]. In this study, as all rice varieties contain the wild-type *EPSPS* gene, conventional PCR using universal primers makes it extremely difficult to detect the mutated *EPSPS* (*mEPSPS*) gene, especially in samples with low *mEPSPS* content, due to overwhelming amplification of the wild-type allele. While mutation-specific PCR can selectively enrich the *mEPSPS* gene, the primers targeting the mutation region prevent sequencing-based confirmation of the exact mutation profile in the amplified product. In contrast, the BDA method overcomes these limitations by suppressing wild-type *EPSPS* amplification, thereby relatively enriching the mutant allele, and using primers outside the mutation region, enabling sequencing of the amplified product to accurately determine the *EPSPS* mutation status.

Currently, qualitative detection methods are the predominant approach in China for testing genetically modified (GM) transformation events [20]. To advance the commercial application of biological breeding and to strengthen the regulatory framework for GM labeling, the Ministry of Agriculture and Rural Affairs issued the “Decision on Revising the Administrative Measures for Agricultural GMO Labeling (Draft for Public Comment)” [21]. According to this regulation, mandatory labeling shall be applied when the GM content exceeds 3% of the product composition for crops listed in the administrative catalog of agricultural GMOs. This policy shift underscores the growing imperative for quantitative detection methodologies in the field.

Rundao118 is a glyphosate-resistant rice variety developed by Sichuan Tianyu Xinghe Biotechnology Co., Ltd. that features mutated *EPSPS* genes in addition to the endogenous wild-type allele [12]. As this material contains both endogenous wild-type and mutated *EPSPS* gene copies, conventional qualitative and quantitative methods often fail to distinguish low-frequency mutations or require complex, costly workflows. This study addresses a critical gap in the field by establishing five robust detection methods for *EPSPS* mutations in glyphosate-resistant rice, each tailored to specific research and regulatory needs. Unlike previous studies that focus on single detection techniques [20,21,22], our work provides a comprehensive comparison of Sanger sequencing, NGS, AS-PCR, qPCR, and BDA, evaluating their sensitivities and applications. Notably, the integration of BDA with sequencing enables precise identification of low-frequency mutations—a significant improvement over existing methods, which often lack the sensitivity or specificity for such tasks. Furthermore, our approach not only facilitates rapid screening but also traces mutation origins (natural vs. engineered), offering unparalleled support for safety assessments and intellectual property protection. This multi-method framework sets a new standard for mutation detection in genetically modified crops, aligning with emerging regulatory requirements, such as China’s 3% labeling threshold for genetically modified organisms (GMOs) [21]. The established methods are versatile, catering to diverse scenarios: NGS for high-throughput sequencing, AS-PCR and qPCR for large-scale screening, and BDA for precise mutation characterization. By bridging the gap between research and practical application, this study provides a foundational toolkit for advancing the commercialization and regulation of glyphosate-resistant rice, while also offering insights applicable to other genetically modified crops.

## 2. Materials and Methods

### 2.1. Plant Materials

Fresh leaf samples from the glyphosate-tolerant rice line Rundao118 were collected during the 2024 growing season in Jiangsu province, China. This rice line was developed through the introduction of an exogenous *EPSPS* gene containing eight nucleotide mutations [12]. Consequently, the line carries three copies of the in vitro-mutated *EPSPS* gene (*mEPSPS*) and one copy of the endogenous wild-type *EPSPS* gene. Nipponbare rice, which has one copy of wild-type *EPSPS* and is sensitive to glyphosate, and 10 different variety samples of conventional rice were preserved in our laboratory and served as the negative control.

### 2.2. DNA Extraction and Sample Preparation

DNA was extracted from each sample using an Efficient Plant Genome DNA extraction kit (TIANGEN, Beijing, China; Catalog #: DP350-02), and the quality and concentration of each sample were measured with a Nanodrop ND-2000 spectrophotometer (ThermoFisher Scientific, Waltham, MA, USA). All of the DNA samples were diluted to a concentration of 25 μg/L and then were stored at −20 °C for the subsequent detection.

### 2.3. Sanger Sequencing of PCR Products

The total DNA extracted from Rundao118 was mixed with that of the Nipponbare rice in specific ratios to prepare DNA samples that consisted of 100%, 10%, 1%, 0.1%, 0.01%, and 0% Rundao118 rice DNA. All the samples were repeated in quadruplicate. For Sanger sequencing, we used the primer pair Osep-1020F/Osep-3081R (Table 1). PCR was performed in a 20 μL reaction mixture containing 10 μL Go Taq^®^ GreenMaster Mix (Promega, Fitchburg, WI, USA; Catalog #: M7122), 1 μL of each forward and reverse primer (10 μM), 2 μL DNA template (25 μg/L), and 6 μL ddH_2_O. The conditions were as follows: 4 min at 95 °C; 35 cycles of 30 s at 95 °C, 30 s at 56 °C, and 90 s at 72 °C; 10 min at 72 °C. After the reaction, each PCR product was subjected to 2% agarose gel electrophoresis, and then all of the gel region containing the target band was cut and sequenced directly using the same pair of primers by Beijing Tsingke Biotech Co., Ltd. (Beijing, China).

### 2.4. NGS of PCR Products

Samples with 100%, 10%, 1%, 0.1%, 0.01%, and 0% Rundao118 rice DNA (prepared as described above) were PCR amplified as for Sanger sequencing. All the samples were repeated in triplicate, and then all of the PCR products were used for NGS as described [23]. Briefly, 0.2 μg PCR product per sample was used as input material for the DNA library preparations. A sequencing library was generated for each sample using an NEBNext^®^ UltraTM DNA Library Prep Kit for Illumina (NEB, Ipswich, MA, USA, Catalog #: E7370L), and index codes were added to each sample. The qualified libraries were pooled and sequenced on Illumina platforms with a PE150 strategy by Beijing Sinobiocore Biological Technology Co., Ltd. (Beijing, China). ANNOVAR (Version: 20230811) was used to annotate single-nucleotide polymorphisms based on the gene model annotation files for the rice reference *EPSPS* gene.

### 2.5. AS-PCR

The total DNA extracted from Rundao118 rice was mixed with that of the Nipponbare rice in specific ratios to prepare DNA samples that consisted of 100%, 10%, 1%, 0.1%, 0.05%, and 0% mutant rice DNA. Each sample was assayed in triplicate. The primers Osep-1075F and Osep-m1240R were used for AS-PCR (Table 1, Figure 2), and their positions in *EPSPS* are shown in Figure 1; this amplified region (nucleotides 1096–1235) covers five of the mutated sites. PCR was performed in a 20 μL reaction mixture containing 10 μL Go Taq^®^ GreenMaster Mix (Promega, Fitchburg, WI, USA; Catalog #: M7122), 1 μL of each forward and reverse primer (10 μM/L), 2 μL DNA template (25 μg/L), and 6 μL ddH_2_O. The conditions were as follows: 4 min at 95 °C; 35 cycles of 30 s at 95 °C, 30 s at 56 °C, and 30 s at 72 °C; 10 min at 72 °C. After the reaction, each PCR product was subjected to 2% agarose gel electrophoresis to look for the allele-specific band of interest.

**Table 1 plants-14-02256-t001:** Primers, probes, and blocker probe used in this study.

Method	Gene	Primer	Sequence	**Product (bp)**
Sanger and NGS methods	*EPSPS and mEPSPS*	Osep-1020F	5′-GGTTATTAGGGCACAACAGTGG-3′	2062
		Osep-3081R	5′-GTAGTCAGGACCTTCTTCAACC-3′	
AS-PCR method	*mEPSPS*	Osep-1075F	5′-ACATGCTTGAGGCCCTGAAAGG-3′	166
		Osep-m1240R	5′-GTCAAGGATCGCATTGCAGTCG-3′	
qPCR method	*mEPSPS*	Osep-1075F	5′-ACATGCTTGAGGCCCTGAAAGG-3′	166
		Osep-m1240R	5′-GTCAAGGATCGCATTGCAGTCG-3′	
		Osep-1140P	5′-FAM-GTAGTCGTTGGCTGTGGTGGCAAG-BHQ1-3′	
	*PLD* [24]	KVM159	5′-TGGTGAGCGTTTTGCAGTCT-3′	68
		KVM160	5′-CTGATCCACTAGCAGGAGGTCC-3′	
		TM013	5′-FAM-TGTTGTGCTGCCAATGTGGCCTG-BHQ1-3′	
BDA method	*mEPSPS*	Osep-1020F	5′-GGTTATTAGGGCACAACAGTGG-3′	228
		Osep-1257R	5′-CAGCAGTCACGGCTGCTGTC-3′	
		Osep-1140P	5′-FAM-GTAGTCGTTGGCTGTGGTGGCAAG-BHQ1-3′	
		Osep-1247BR	5′-GGCTGCTGTCAATGGTCGCATTGCAGTTCTTTT-3′	

### 2.6. qPCR

For the qPCR, DNA extracted from the Rundao118 rice leaves was serially diluted with ddH_2_O to prepare five calibration standards with five different copy numbers of *mEPSPS* or internal reference gene phospholipase D (*PLD*) (24,000, 4800, 960, 192, 38.4). These calibration standards were then used as templates for the qPCR reactions to generate a standard curve. Meanwhile, we prepared a sample with a 3% concentration of Rundao118 DNA as described above to validate the accuracy of this quantitative method. All the samples were assayed in triplicate. We again used the primer pair Osep-1075F/Osep-m1240R, and the probe Osep-1140P was used to amplify *mEPSPS* (Table 1, Figure 2). The primer pair KVM159/KVM160 and the probe TM013 were used to amplify *PLD* (Table 1) [24].

PCR was performed in a 20 μL reaction mixture containing 10 μL Premix Ex Taq™ Mix (TaKaRa Bio, Dalian, China; Catalog #: RR390B), 1 μL each of the forward and reverse primers (10 μM), 0.5 μL probe (10 μM), 2 μL DNA template (25 μg/L), and 5.5 μL ddH_2_O. The conditions were as follows: 20 s at 95 °C and 40 cycles of 5 s at 95 °C and 34 s at 60 °C. Then, the fluorescence signals were collected at 60 °C. qPCR was carried out with an ABI7500 real-time PCR machine (Applied Biosystems, Foster City, CA, USA), and the data were analyzed with 7500 Software v2.0 Software (Applied Biosystems). The copies of *mEPSPS* and the internal reference gene *PLD* in each sample were calculated based on their Ct values [18], and the content of Rundao118 rice was calculated according to the following equation:Content of Rundao118 rice (%) = copies of *mEPSPS*/copies of *PLD* × 100%.

### 2.7. BDA

For BDA, six different samples consisting of different Rundao118 DNA percentages (100%, 10%, 1%, 0.1%, 0.01%, and 0%) were amplified; meanwhile, 10 different conventional variety rice samples were used to validate the specificity and effectiveness of this BDA method. All the samples were assayed twice. The primers designed for this method (Osep-1020F and Osep1257R) target five of the mutation sites in nucleotide region 1096–1235, with the upper and lower primers being located outside the mutation region (Figure 2). The probe Osep-1140P is the same as that used with the qPCR method, and the blocker probe Osep-1247BR is located in nucleotide region 1219–1235, which is consistent with the wild-type gene sequence (Table 1).

BDA was carried out in a 20 µL reaction volume containing 10 μL Premix Ex Taq™ Mix (TaKaRa Bio, Dalian, China; Catalog #: RR390B), 1 μL each of the forward and reverse primers (10 μM/L), 0.5 μL probe (10 μM), 5 μL blocker probe (10 μM), 2 μL template DNA (25 μg/L), and 0.5 μL ddH_2_O. The conditions were as follows: 20 s at 95 °C, 40 cycles of 5 s at 95 °C and 34 s at 60 °C. Then, the fluorescence signals were collected at 60 °C. The samples to be tested need to be controlled without blockers, and all of the enriched PCR products were sequenced by Sanger sequencing using primer Osep-1020F.

## 3. Results

### 3.1. Sanger Method

The gel electrophoresis results from this method show a 2062 bp band, which is consistent with the expected target fragment size (Figure 3A). Subsequent Sanger sequencing revealed eight nucleotide alterations in the *EPSPS* sequence of Rundao118 as compared with the control Nipponbare. Specifically, there was a C→G at position 1096, a G→C at position 1219, an A→G at position 1220, a C→T at position 1233, an A→C at position 1235, an A→G at position 1883, an A→C at position 1993, and a T→C at position 2579 (Figure 3B). The sequencing chromatograms displayed distinct double peaks in both 100% and 10% Rundao118 concentration samples (Figure 3C,D), whereas only the non-mutated nucleotide peaks were observed in the 1%, 0.1%, 0.01%, and 0% samples (Figure 3E–H).

### 3.2. NGS Method

We also subjected the same PCR amplification products from the Sanger method to NGS sequencing. Samples for all six concentrations of Rundao118 DNA yielded > 0.79 Gb of data, corresponding to a coverage depth exceeding 250,000×. The results demonstrate consistent mutation frequencies at all eight sites within samples of identical concentrations. The mean combined mutation rates (with all eight sites simultaneously mutated) were 78.11% for the 100% sample, 23.33% for the 10% sample, 3.24% for the 1% sample, 0.67% for the 0.1% sample, 0.53% for the 0.01% sample, and 0.21% for the 0% sample (Table 2). Given that this material contains three copies of *mEPSPS* and one copy of endogenous *EPSPS*, the theoretical mutation rates were calculated as 75.00%, 23.08%, 2.91%, 0.30%, 0.03%, and 0.00% for the respective concentrations. These data indicate that NGS maintains a high accuracy for samples above a 1% mutation frequency. Additionally, the 0% sample revealed an NGS background noise level of ~0.2%.

### 3.3. AS-PCR Method

As shown in Figure 4, PCR amplification using DNA templates containing 0.05% Rundao118 rice DNA or higher consistently yielded the expected 166 bp specific band (lanes 1–10). In contrast, neither the control Nipponbare rice samples (lanes 16–18) nor the blank controls (lanes 19–21) produced the target amplicon. Thus, this PCR method exhibits excellent specificity, with a detection sensitivity reaching 0.05% mutated DNA.

### 3.4. qPCR Method

Figure 5A,B demonstrate that the standard curves for *mEPSPS* and for the internal reference *PLD* had slopes of −3.3615 and −3.3421, with R^2^ values of 0.9990 and 0.9975, respectively. These results meet the requirements for qPCR analysis (−3.6 ≤ slope ≤ −3.1, R^2^ ≥ 0.98), indicating that the qPCR method exhibits excellent specificity and a detection sensitivity reaching 38.4 copies. Furthermore, the validity of this method was verified using a sample with 3% Rundao118 DNA, with the measured value, 2.97 ± 0.26%, closely matching the theoretical value of 3% (Table 3).

### 3.5. BDA Method

For rice DNA samples containing different concentrations of Rundao118 DNA, all the samples exhibited robust amplification curves in the control reaction system without the suppression probe Osep-1247BR. However, except for the 100% sample, the remaining concentrations could not be effectively distinguished (Figure 6A). In contrast, in the BDA reaction system with the suppression probe, samples with concentrations of 0.1% and above showed specific and well-defined amplification curves, displaying typical gradient characteristics (Figure 6B). Among the 10 conventional rice samples that did not contain *mEPSPS*, amplification was significantly delayed when the blocking probe was added, which prevented a typical amplification curve from forming. In contrast, typical amplification curves were observed when the suppression probe was omitted (Figure 6C).

The amplification products from the rice DNA samples containing different concentrations of the Rundao118 DNA were analyzed by agarose gel electrophoresis. All the samples produced a distinct amplification band of 228 bp in the absence of the blocking probe. When the blocking probe was included, samples at concentrations above 0.1% also generated the characteristic 228 bp band (Figure 6D).

In the control system without the blocking probe, the sequencing chromatograms of the amplification products (Figure 6E–G, −B) showed mixed peaks at the five mutation sites (positions 1096, 1219, 1220, 1233, and 1235) for the 100% and 10% sample, faint mutation peaks for the 1% sample, and no detectable mutation peaks for the 0.1% sample. Consequently, the actual mutant sequences in these samples could not be definitively determined. In contrast, the chromatograms from the BDA system with the blocking probe (Figure 6E–H, +B) showed that the amplification products of DNA samples containing 0.1% or higher concentrations of *mEPSPS* DNA exhibited *mEPSPS* gene sequences exclusively across all five mutation sites in the amplified region (Figure 6E–H).

### 3.6. Comparison of the Five Detection Methods

We compared five detection methods developed in current study across several key parameters (Table 4). The methods employed either conventional PCR thermocyclers or a real-time PCR system, demonstrating varying capabilities in mutation detection. The sensitivity differed significantly, with qPCR achieving the highest detection limit (0.01%), followed by BDA (0.1%), AS-PCR (0.05%), NGS (1%), and Sanger (10%). The turnaround time ranged from 2 to 4 h for rapid screening methods (qPCR) from 24 to 72 h for comprehensive analyses (Sanger, NGS). The methods exhibited distinct application profiles based on their performance characteristics. Sanger, NGS, and BDA are optimal for precise mutation characterization, with BDA specifically noted for low-frequency mutation detection. AS-PCR and qPCR excel in large-scale, rapid screening of known mutation sites, offering high throughput and speed.

## 4. Discussion

The development of glyphosate-resistant rice through targeted mutations in the *EPSPS* gene represents a significant advancement in agricultural biotechnology [25]. Our study establishes five distinct methods for detecting *EPSPS* mutations in the glyphosate-resistant rice line Rundao118, each with unique advantages in sensitivity, specificity, and applicability. In this study, we initially used the Sanger sequencing method to determine the sequences of PCR products to characterize a rice strain for the presence of glyphosate-resistant mutations in *EPSPS*. Sequence alignment analysis revealed eight base variations in *EPSPS* from the mutant Rundao118 rice strain as compared with that from the control Nipponbare strain. These variant sites were completely consistent with the sequence changes reported in the patent for Rundao118 [12]. Although natural variations in the *EPSPS* sequence exist in nature, the simultaneous detection of multiple reported characteristic mutation sites associated with artificial genetic manipulation in an unknown material allows us to reasonably conclude that such mutations originate from artificial genetic manipulation rather than natural variation. Through sequencing chromatogram analysis of samples with varying concentrations of Rundao118 DNA relative to Nipponbare DNA, we observed distinct double peaks in the 10% and 100% concentration samples, whereas we detected only non-mutated nucleotide peaks in the 1%, 0.1%, 0.01%, and 0.00% samples. This phenomenon can be explained as follows: As the test material contains both exogenous insertions and endogenous *EPSPS* genes, exogenous *mEPSPS* fails to amplify effectively at low concentrations, resulting in only non-mutated nucleotide peaks in the 1%, 0.1%, and 0.01% samples. Based on these experimental results, we speculate that the detection sensitivity of this method is ~10%.

The NGS sequencing results show a mutation rate of 0.21% in the 0% concentration sample, which aligns with the typical systematic error range (0.1%~0.3%) of NGS [26,27,28]. Therefore, we subtracted this 0.21% as background noise as a correction. After adjustment, the mutation rates of the 100%, 10%, 1%, 0.1%, and 0.01% concentration samples were 77.90%, 23.02%, 3.03%, 0.46%, and 0.32%, respectively. Given that *EPSPS* in this material consists of three exogenous insertion copies and one endogenous copy, the theoretical mutation rates for the respective concentration samples should be 75.00%, 23.08%, 2.91%, 0.30%, 0.03%, and 0.00%. The experimental results demonstrate strong agreement between the measured and theoretical values for the 100%, 10%, and 1% concentration samples. However, slight deviations were observed in the 0.1% and 0.01% low-concentration samples. Based on these findings, we determined that the detection sensitivity of this method is ~1%. Notably, based on the mutation rate data, we calculated that the copy numbers of the exogenous insertion sequence in the 100%, 10%, and 1% concentration samples were 3.57, 3.04, and 3.35, respectively. These results show strong consistency with the actual copy number (three copies) of *mEPSPS*, further validating the reliability of the experimental findings. This discovery demonstrates that even when the copy number of exogenous insertion sequences is unknown, NGS sequencing can still provide preliminary estimates, offering valuable reference data for subsequent analyses.

Our results demonstrate that Sanger sequencing, while reliable for identifying mutations at a sensitivity of 10%, is limited by its lower throughput compared to NGS. This aligns with previous studies highlighting NGS as a superior alternative for high-throughput mutation detection, offering a sensitivity of 1% and the ability to analyze multiple samples simultaneously [13]. The NGS data also allowed us to estimate the copy number of exogenous *mEPSPS*, a feature particularly valuable for characterizing genetically modified (GM) organisms with unknown insertion events. Similar approaches have been employed in other systems to quantify transgene copy numbers [29], underscoring the versatility of NGS in genetic research.

Currently, qualitative detection methods are typically used for detecting genetically modified organism (GMO) events in China [20]. However, with the rapid advancement of bio-breeding industrialization [25], coupled with the revision of the Agricultural GMO Labeling Administration Measures and the establishment of a 3% labeling threshold [21], quantitative detection technologies are inevitably becoming more popular. To address this need, we have established two detection methods (AS-PCR and qPCR) for *EPSPS* characterization. In the field of GMO quantitative detection, qPCR and ddPCR serve as the two primary technical approaches. Specifically, qPCR achieves relative quantification through standard curve construction [17,18]. In contrast, ddPCR enables absolute quantification without reliance on standard curves. Despite its notable advantages, the widespread adoption of ddPCR remains constrained by high instrumentation costs and expensive consumables [30,31]. The results from this study indicate that both the established AS-PCR and qPCR methods exhibit excellent specificity, with detection sensitivities reaching 0.05% and 0.01%, respectively. These performance metrics fully meet the current requirements for GMO detection and provide reliable technical support for the accurate identification of GM products.

Furthermore, this study successfully developed a novel BDA-based detection method. While BDA technology has been primarily utilized in clinical diagnostics, its application in plant biotechnology has been limited due to several technical challenges. The conventional BDA approach is prone to nonspecific amplification, primer-dimer formation, and uneven amplification efficiency when excessive primers and probes are employed, which can substantially compromise reaction stability and detection accuracy. Furthermore, the technique’s performance is sensitive to minor variations in reaction conditions, including temperature fluctuations and salt concentration changes, which may adversely affect hybridization efficiency and consequently impact assay sensitivity and specificity [32,33]. Through systematic optimization of probe design and reaction conditions, we successfully minimized nonspecific amplification, thereby overcoming a major limitation inherent to conventional BDA techniques [14]. By coupling BDA amplification with sequencing analysis, our optimized method enables precise identification of point mutations within the *EPSPS* nucleotide region (1096–1235), achieving exceptional sensitivity with a detection limit of 0.1% mutant allele frequency in complex DNA samples. This integrated approach not only maintains high sensitivity but also ensures accurate mutation characterization at single-nucleotide resolution. The developed BDA method exhibits remarkable advantages for allele-specific enrichment in plant biotechnology applications. Its unique capability to selectively suppress wild-type *EPSPS* amplification while enriching mutant alleles addresses a critical challenge documented in previous studies [34,35], particularly for detecting low-frequency mutations in genetically heterogeneous samples. With its single-nucleotide discrimination power, this BDA-based platform represents a significant advancement for characterizing gene-edited crops, offering unprecedented precision in mutation profiling as required by cutting-edge applications described in recent literature [9,11]. The achieved 0.1% detection sensitivity substantially enhances our ability to identify and analyze rare mutations in complex genomic backgrounds, providing a powerful tool for quality control and regulatory assessment of gene-edited crops.

Based on the methodology established in this study, we propose expanding the detection regions to enable comprehensive coverage of target genes, or even multiple potential mutant variants. This advancement will facilitate development of a high-throughput system for direct detection of genetically modified (GM) traces in paddy field environments. Our detection platform addresses a critical gap in the current monitoring capabilities for glyphosate-resistant crops, particularly given increasingly stringent regulatory requirements. For example, China’s proposed 3% GMO labeling threshold [21]—validated by our qPCR results—demands highly sensitive quantification methods. Similar regulatory frameworks in the EU, U.S. [36], and other regions highlight the global significance of this work. Importantly, our approach’s capacity to distinguish mutation origins (natural variation vs. gene editing vs. transgenesis) provides valuable tools for both intellectual property protection and biosafety evaluation.

While our methods exhibit high sensitivity, challenges remain. For example, NGS requires substantial computational resources, and the BDA method relies on carefully optimized primer design for robust performance. Future research could explore multiplexing strategies or CRISPR-based detection systems to further improve efficiency. Additionally, expanding these methods to other crops and herbicide-resistance genes would broaden their applicability.

## 5. Conclusions

In summary, our study establishes five methods for detecting *EPSPS* mutations in glyphosate-resistant rice with the following sensitivities: Sanger sequencing (10%), NGS (1%), AS-PCR (0.05%), qPCR (0.01%), and BDA (0.1%). Sanger, NGS, and BDA provided single-nucleotide resolution for precise mutation characterization, while AS-PCR and qPCR enabled efficient large-scale screening. The BDA method, in particular, combined high sensitivity with the ability to suppress wild-type amplification, facilitating accurate low-frequency mutation detection. These methods collectively address critical gaps in glyphosate-resistant rice analysis, from basic research to regulatory compliance. Their complementary sensitivities and applications provide researchers with flexible tools for mutation identification, quantification, and origin tracing. Future work could explore integrating these techniques for enhanced detection workflows in plant biotechnology.

## Figures and Tables

**Figure 1 plants-14-02256-f001:**
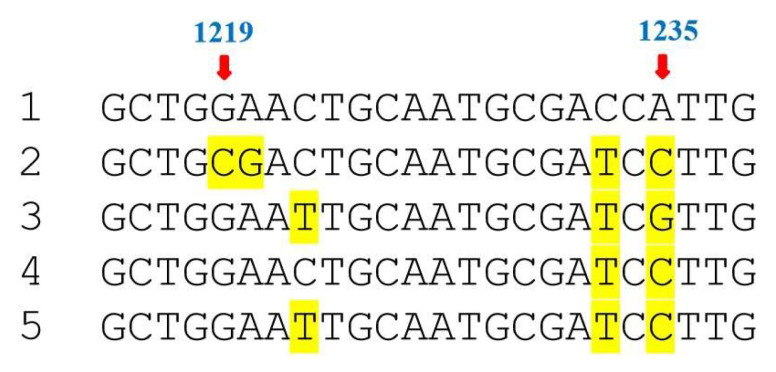
Sequence comparison of key nucleotide mutations in *EPSPS* genes between wild-type and modified variants. 1: Wild-type *EPSPS* gene from Nipponbare rice (control); 2: Mutated *EPSPS* developed by Sichuan Tianyu Xinghe Biotechnology Co., Ltd. [12]; 3: Modified *EPSPS* sequence reported by Gao et al. [9]; 4: Engineered *EPSPS* variant generated by Xia et al. [11]; 5: Mutated *EPSPS* gene construct described by Luo et al. [8]. Red arrows indicate mutation locations. The yellow highlights show nucleotide mutations compared to the wild-type *EPSPS* gene.

**Figure 2 plants-14-02256-f002:**
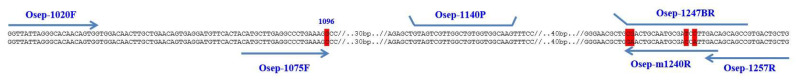
The position of primers, probes, and blocker probe used for the AS-PCR, qPCR, and BDA methods for *EPSPS* amplification. Primers used for the AS-PCR method: Osep-1075F/Osep-m1240R. Primers and probe used for the qPCR method: Osep-1075F/Osep-m1240R and probe Osep-1140P. Primers, probe, and blocker probe used for the BDA method: Osep-1020F/Osep-1257R, probe Osep-1140P, and blocker probe Osep-1247BR. The upper sequence shows the mutated *EPSPS* sequence; the lower sequence shows the wild-type *EPSPS* sequence. The red highlights show nucleotide mutations compared to the wild-type *EPSPS* gene.

**Figure 3 plants-14-02256-f003:**
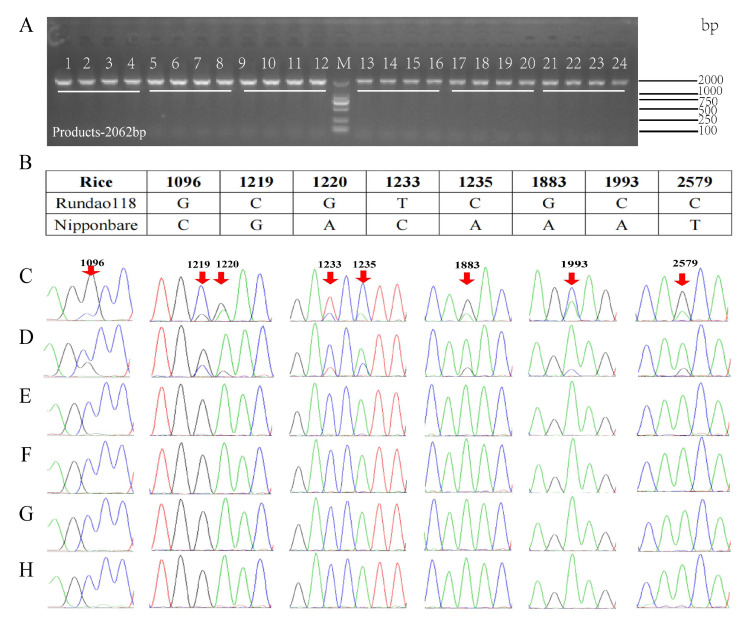
Detection results by Sanger sequencing. (**A**) Electrophoresis profile of PCR products. All samples were analyzed in quadruplicate. Lanes 1–4: 100% Rundao118 DNA; 5–8: 10% Rundao118 DNA; 9–12: 1% Rundao118 DNA; 13–16: 0.1% Rundao118 DNA; 17–20: 0.01% Rundao118 DNA; 21–24: 0% Rundao118 DNA; M: DL2000 marker. (**B**) Nucleotide variation at eight *EPSPS* loci between RunDao118 and Nipponbare rice plants. (**C**–**H**) Sanger sequencing chromatograms for samples with 100% (**C**), 10% (**D**), 1% (**E**), 0.1% (**F**), 0.01% (**G**), and 0% (**H**) Rundao118 DNA concentrations, respectively. Red arrows indicate mutation sites.

**Figure 4 plants-14-02256-f004:**
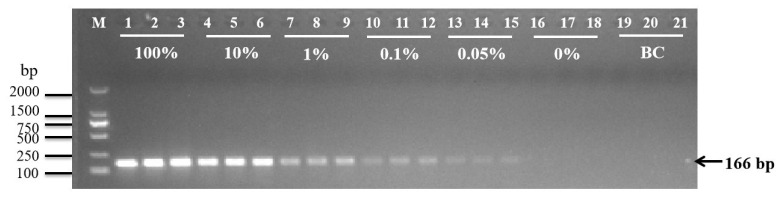
Electrophoresis results from the AS-PCR method for samples with different concentrations of Rundao118. Lanes 1–3: 100%; 4–6: 10%; 7–9: 1%; 10–12: 0.1%; 13–15: 0.05%; 16–18: 0%; 19–21: blank control (BC). The negative control used was 0% Rundao118 DNA.

**Figure 5 plants-14-02256-f005:**
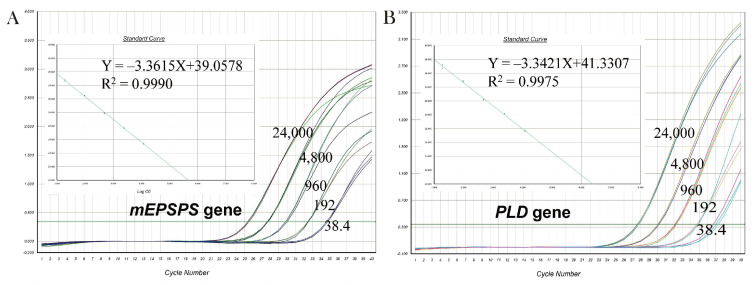
qPCR detection results for *mEPSPS*. (**A**) Amplification curve and standard curve for *mEPSPS*. (**B**) Amplification profile and standard curve for the rice endogenous reference gene *PLD*. Notes: Numbers on the amplification curves indicate copy numbers.

**Figure 6 plants-14-02256-f006:**
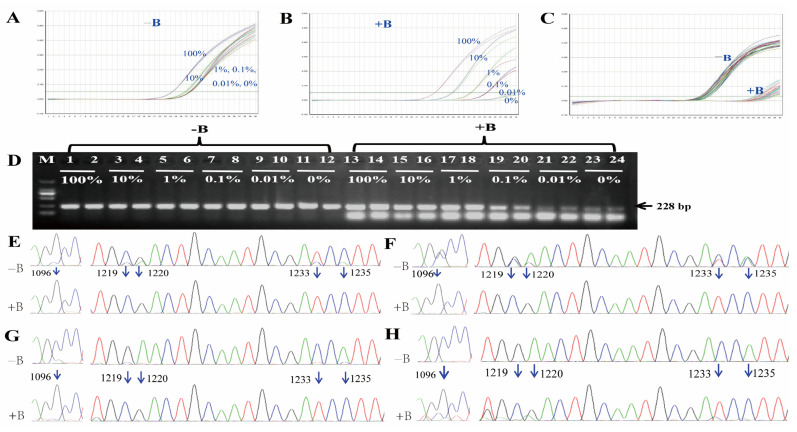
Detection results for the BDA method. (**A**) Amplification curve for *mEPSPS* for different concentrations of Rundao118 DNA in the samples without the blocking probe (−B). (**B**) Amplification curve for *mEPSPS* for different concentrations of Rundao118 DNA in the samples with the blocking probe (+B). (**C**) Amplification curve of the *EPSPS* gene for DNA samples from 10 conventional rice varieties. (**D**) Electrophoresis results for *EPSPS* as the product of BDA. (**E**–**H**) Sanger sequencing chromatograms for samples with 100% (**E**), 10% (**F**), 1% (**G**), and 0.1% (**H**) Rundao118 DNA. Blue arrows indicate mutation sites.

**Table 2 plants-14-02256-t002:** Sequencing results of NGS method for samples with different concentrations of Rundao118.

Sample	Rep.	Clean Base (Gb) *	1096 (G/N ^†^, %)	1219 (C/N ^†^, %)	1220 (G/N ^†^, %)	1233 (T/N ^†^, %)	1235 (C/N ^†^, %)	1883 (G/N ^†^, %)	1993 (C/N ^†^, %)	2579 (C/N ^†^, %)	Average (%)	Test Value (Mean ± SE, %)	Theoretical Value (%) ^#^
S1 (100%)	1	0.79	77.18	75.87	76.16	75.76	76.35	78.87	78.44	78.62	77.16	78.11 ± 1.05	75.00
	2	0.96	79.27	78.03	78.34	77.83	78.41	80.92	80.52	80.61	79.24		
	3	0.96	77.82	76.57	77.01	76.51	77.15	79.63	79.22	79.52	77.93		
S2 (10%)	1	0.96	22.71	21.96	22.27	22.00	22.19	23.11	22.71	22.92	22.48	23.33 ± 1.06	23.08
	2	1.46	24.76	23.91	24.28	24.04	24.14	25.24	24.84	24.93	24.52		
	3	1.21	23.27	22.47	22.77	22.52	22.72	23.58	23.06	23.53	22.99		
S3 (1%)	1	1.05	3.14	2.95	3.29	3.01	3.08	3.25	3.00	3.24	3.12	3.24 ± 0.19	2.91
	2	1.07	3.41	3.23	3.52	3.30	3.37	3.75	3.45	3.69	3.47		
	3	1.26	3.08	2.94	3.29	3.02	3.05	3.35	3.16	3.29	3.15		
S4 (0.1%)	1	0.92	0.74	0.71	1.01	0.80	0.90	0.98	0.76	0.90	0.85	0.67 ± 0.16	0.30
	2	1.15	0.48	0.44	0.76	0.53	0.62	0.71	0.46	0.61	0.58		
	3	1.19	0.50	0.46	0.74	0.52	0.59	0.64	0.49	0.64	0.57		
S5 (0.01%)	1	0.97	0.75	0.69	1.03	0.76	0.84	0.86	0.66	0.90	0.81	0.53 ± 0.25	0.03
	2	0.89	0.35	0.33	0.70	0.40	0.45	0.51	0.32	0.49	0.44		
	3	1.32	0.23	0.22	0.50	0.34	0.33	0.47	0.21	0.45	0.34		
S6 (0%)	1	1.21	0.11	0.09	0.43	0.18	0.18	0.32	0.10	0.28	0.21	0.21 ± 0.02	0.00
	2	1.22	0.08	0.06	0.40	0.15	0.21	0.28	0.08	0.27	0.19		
	3	1.34	0.13	0.11	0.41	0.18	0.18	0.36	0.14	0.33	0.23		

* Gb: gigabases (1 × 10^9^ base pairs). ^†^ N: all nucleotides (A, T, G, and C) that were sequenced. ^#^ Theoretical Value (%) = (3 × mutant rice DNA concentration)/(3 × mutant rice DNA concentration + 1) × 100%.

**Table 3 plants-14-02256-t003:** qPCR verification test results with 3% Rundao118 DNA samples.

Rep.	Gene	Ct Value			Copy Number	Content (%)	Average Content * (%)
1	*mEPSPS*	30.31	30.34	30.37	408.24	2.87	2.97 ± 0.26
	*PLD*	27.06	27.07	27.14	14,239.20		
2	*mEPSPS*	30.08	30.06	30.06	414.81	2.78	
	*PLD*	26.46	26.51	26.47	14,903.10		
3	*mEPSPS*	30.04	29.93	30.09	457.11	3.26	
	*PLD*	26.38	26.22	26.33	14,002.80		

* Shown as mean ± SE.

**Table 4 plants-14-02256-t004:** Comparison of the five detection methods developed in this study.

Methods	Detection Instrument Requirements	Target Region (Number of Mutation Sites)	Sensitivity (%)	Time (h)	Applicable Scenarios
Sanger	Conventional PCR Thermocycle	1020–3081 (8)	10%	24~48	precise mutation site characterization and identification
NGS	Conventional PCR Thermocycle	1020–3081 (8)	1%	~72	precise mutation site characterization and identification
AS-PCR	Conventional PCR Thermocycle	1075–1240 (5)	0.05%	3~6	large-scale rapid screening of known mutation sites
qPCR	Real-time PCR System	1075–1240 (5)	0.01%	2~4	large-scale rapid screening of known mutation sites
BDA	Real-time PCR System	1020–1257 (5)	0.1%	3~5	precise mutation site characterization and identification, especially low-frequency mutations

## Data Availability

All the data generated or analyzed during this study can be found in the article; further inquiries can be directed to the corresponding author.

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
