# Peer review of "Methods Established for EPSPS Gene Mutation Detection in Glyphosate-Resistant Rice (Oryza sativa L.)"

_plants, 2025, doi:10.3390/plants14152256_

Round 1
Reviewer 1 Report
Comments and Suggestions for Authors
This manuscript successfully established five methods targeted to EP-SPS specifically for the qualitative or quantitative detection of Rundao118 rice that its genome contains three copies of exogenous mutated EP-SPS and one copy of wild type ES-SPS. Beyond this, these methods facilitate the rapid determination of mutated EPSPS sequences in Rundao118 rice, the distinguishment of mutation from natural or artificial modification, enriching the arsenal of tools for genetically modified organism (GMO) detection and intellectual property protection. The manuscript provides solid results and is well written.
Minor issues
Fig 1, add the location of the primer Osep-3081R into the diagram.
Table 2, describe clearly how the theoretical value (%) was calculated.
Compare lanes 21-24 in Fig 5D and lanes 21-24 in Fig 2A, why the intensity of the band looked so different? Any explanation?
Author Response
REVIEWER 1: Minor issues
C1: Fig 1, add the location of the primer Osep-3081R into the diagram.
R1: The primer Osep-3081R is located at position 3081 of the rice EPSPS gene, which is nearly 2000 bp downstream of the region shown in Figure 1, making it impractical to display in the figure.
C2: Table 2, describe clearly how the theoretical value (%) was calculated.
R2: the theoretical value (%) = (3 × mutant rice DNA concentration)/(3 × mutant rice DNA concentration+1)×100%.
We have added the above information in the revised manuscript, please see Table 2 marked in blue.
C3: Compare lanes 21-24 in Fig 5D and lanes 21-24 in Fig 2A, why the intensity of the band looked so different? Any explanation?
R3: In Figure 2A, all lanes contain samples co-amplifying both mutated and non-mutated EPSPS genes, with nearly identical target concentrations across samples. As a result, no significant differences in amplification intensity are observed between lanes. In contrast, in Figure 5D (lanes 21–24), only the mutated EPSPS gene is amplified. Due to the extremely low concentration of the target template, the amplification bands appear faint.

Reviewer 2 Report
Comments and Suggestions for Authors
(1) Please write the Latin name of the rice (Oryza L.) or (Oryza sativa L.) in the title.
(2) Please check the websites (lines 45 and 93). The content is little understood (no English translation available). The second website does not work. You might want to include the web pages in the bibliography as well.
(3) Please indicate the validity of the sequencing or modify and expand the information in the paragraph (lines 81-87). In the introduction it is described that the sequencing only detects the wild-type sequence (non-mutated ) of EPSPS. In contrast, the results say otherwise.
(4) Materials and methods: please specify how many samples were used. 1 resistant variety (in analyses with different concentrations of the mutated DNA), 1 sample of control and 10 conventional varieties? Is there a different number of samples in each analysis. It is easy to get lost on how many samples and which ones were used in each experiment.
(5) The materials and methods do not describe the use of different concentrations of mutant DNA for sequencing, whereas the results do. This is worth noting in the description of the sequencing methodology.
(6) There is a row in Table 1 with PLD, which was not mentioned anywhere in the methodology. It is worth adding this information.
(7) Why did AS-PCR use different concentrations of mutant DNA than the other methods? AS-PCR (100%, 10%, 1%, 0.1%, 0.05% and 0%), other methods (100%, 10%, 1%, 0.1%, 0.01% and 0%).
(8) Discussion: this chapter presents the summary results and conclusions. No discussion is provided. There are no references to other papers on similar topics and no discussion of the results in comparison with other scientific papers. Consequently, I can not classify this chapter as a “discussion”. This chapter needs significant improvement.
(9) References: please add more recent articles to your knowledge. Only 5 references are from the last 5 years.
Author Response
C1: Please write the Latin name of the rice (Oryza L.) or (Oryza sativa L.) in the title.
R1: Done as suggested, please see lines 3 marked in blue.
C2: Please check the websites (lines 45 and 93). The content is little understood (no English translation available). The second website does not work. You might want to include the web pages in the bibliography as well.
R2: We have changed the websites into References. Please see lines 46 and 96.
C3: Please indicate the validity of the sequencing or modify and expand the information in the paragraph (lines 81-87). In the introduction it is described that the sequencing only detects the wild-type sequence (non-mutated ) of EPSPS. In contrast, the results say otherwise.
R3: We have rephrased the paragraph, please see lines 81-90 marked in blue.
C4: Materials and methods: please specify how many samples were used. 1 resistant variety (in analyses with different concentrations of the mutated DNA), 1 sample of control and 10 conventional varieties? Is there a different number of samples in each analysis. It is easy to get lost on how many samples and which ones were used in each experiment.
R4: Done as suggested, please see Materials and methods section, lines 143, 155-156, 168, 185, 214 marked in blue.
C5: The materials and methods do not describe the use of different concentrations of mutant DNA for sequencing, whereas the results do. This is worth noting in the description of the sequencing methodology.
R5: Done as suggested, please see Materials and methods section, lines 150-151, 156, 225-226 marked in blue.
C6: There is a row in Table 1 with PLD, which was not mentioned anywhere in the methodology. It is worth adding this information.
R6: the PLD was mentioned in the section 2.6. qPCR. We have added the full name of PLD in line 181 marked in blue, and reference in line 188 and in the table 1 marked in blue.
C7: Why did AS-PCR use different concentrations of mutant DNA than the other methods? AS-PCR (100%, 10%, 1%, 0.1%, 0.05% and 0%), other methods (100%, 10%, 1%, 0.1%, 0.01% and 0%).
R7: Since the detection limit in qualitative PCR for genetically modified rice is generally 0.05%, samples at this concentration were included in the AS-PCR qualitative PCR method used in this study.
C8: Discussion: this chapter presents the summary results and conclusions. No discussion is provided. There are no references to other papers on similar topics and no discussion of the results in comparison with other scientific papers. Consequently, I can not classify this chapter as a “discussion”. This chapter needs significant improvement.
R8: Done as suggested, please see Discussion section marked in blue.
C9: References: please add more recent articles to your knowledge. Only 5 references are from the last 5 years.
R9: Done as suggested, please see Reference section marked in blue.
Reviewer 3 Report
Comments and Suggestions for Authors
Dear respected authors,
I have carefully read your article. Very impressive and well-done work has been done in this paper. Please consider the following comments during revision and then edit the article accordingly:
- Prepare a flowchart for the Materials and Methods section and summarize the conducted steps. You can use software like BioRender for this purpose.
- There are grammatical and language errors in some parts of the article. Please carefully proofread the text.
- Prepare a list of abbreviations and include all abbreviated terms and expressions at the end of the manuscript.
- Mention the significance and novelty of this article in comparison with other publications in this field in the Introduction section.
- The authors have used five major methods to identify transgenic rice lines resistant to glyphosate herbicide. However, in each group of evaluation methods, the control treatments used for result comparison have not been clearly explained. Please provide more details about the control treatments.
- Lines 32 to 36: To illustrate the amino acid changes in the structure of the EPSPS gene and protein in rice compared to other resistant species, you can use a nucleotide or protein BLAST. This analysis can effectively show the differences in the amino acid structure of this protein and similar ones.
- Line 45: The provided link refers to a Chinese website and the information is not accessible. Please use a more reliable source that is available in English.
- Lines 46–53: Please include one or two references in this section. Additionally, some of the discussed paragraphs in other parts of the article lack citations. Please use credible references to support the content in each section of the article.
- Lines 46–80: You can illustrate the advantages and disadvantages of each mentioned method in this section using a well-designed graphical figure. This would help reduce the volume of text and enhance the article's visual appeal and quality.
- Lines 88–96: Please cite the appropriate references in this section.
- Unavailable web address: The address https://www.moa.gov.cn/hd/zqyj/index_1.htm is not accessible. If possible, please provide a screenshot of the content used from this site and include it in the article’s supplementary file.
- Lines 111–114: Please include references for this section. If the transgenic rice mentioned was developed by the authors, more detailed information should be provided in the article’s supplementary file. Otherwise, supporting documents—such as publications related to the development of this resistant line and other relevant data—must be properly cited in the manuscript.
- Line 112: Please include the catalog number of the kit used in this section. Also, mention the catalog numbers of all other kits used throughout the study.
- Please provide references for all protocols mentioned in the Materials and Methods section, especially the original articles or manuals from which the protocols were derived.
- Figure 2: Clearly indicate the control group in the figure. The article should include either positive or negative controls to properly interpret the experimental results.
- Line 190: The format of the cited reference in this line does not match the format of other in-text citations. Please ensure that all references follow a consistent format and that a DOI (Digital Object Identifier) is included for each cited paper.
- Please include a general discussion in the Discussion section about the industrial applicability of this work and its results in monitoring rice fields globally.
- In the Discussion section, compare the limitations of the methods used in this study.
- The conclusion section is too long and includes repetitive content from other sections of the manuscript. Please rewrite this section to concisely summarize only the key experimental findings.
- The references used in the article are outdated. Please use more recent and up-to-date references to support the discussion and compare the obtained results with those reported in recently published articles from scientific databases.
Author Response
C1: Prepare a flowchart for the Materials and Methods section and summarize the conducted steps. You can use software like BioRender for this purpose.
R1: We appreciate the reviewer's suggestion to include a flowchart for the Materials and Methods section. While we agree that visual aids can enhance clarity, we believe the current text-based description already provides a comprehensive and logically structured account of our experimental procedures. Furthermore, given that the manuscript has already reached 15 pages, the inclusion of a flowchart would make the document excessively long. We fully recognize the merit of this suggestion for enhancing methodological transparency, particularly for complex experimental designs. We will definitely implement this approach in our future large-scale studies.
C2: There are grammatical and language errors in some parts of the article. Please carefully proofread the text.
R2: Done as suggested, please read through the whole manuscript.
C3: Prepare a list of abbreviations and include all abbreviated terms and expressions at the end of the manuscript.
R3: Done as suggested, please see the Abbreviated section, lines 479-483 marked in blue.
C4: Mention the significance and novelty of this article in comparison with other publications in this field in the Introduction section.
R4: Done as suggested, please see the Introduction section, lines 103-123 marked in blue.
C5: The authors have used five major methods to identify transgenic rice lines resistant to glyphosate herbicide. However, in each group of evaluation methods, the control treatments used for result comparison have not been clearly explained. Please provide more details about the control treatments.
R5: Done as suggested, please see Materials and methods section, lines 130-133, 277 marked in blue.
C6: Lines 32 to 36: To illustrate the amino acid changes in the structure of the EPSPS gene and protein in rice compared to other resistant species, you can use a nucleotide or protein BLAST. This analysis can effectively show the differences in the amino acid structure of this protein and similar ones.
R6: Done as suggested, please see Introduction section, lines 46-53 marked in blue.
C7: Line 45: The provided link refers to a Chinese website and the information is not accessible. Please use a more reliable source that is available in English.
R7: We have changed the websites into References. Please see lines 46 and 96.
C8: Lines 46–53: Please include one or two references in this section. Additionally, some of the discussed paragraphs in other parts of the article lack citations. Please use credible references to support the content in each section of the article.
R8: Done as suggested, please see lines 57, 60, and Reference section marked in blue.
C9: Lines 46–80: You can illustrate the advantages and disadvantages of each mentioned method in this section using a well-designed graphical figure. This would help reduce the volume of text and enhance the article's visual appeal and quality.
R9: We have incorporated a comprehensive comparison table (Table 4) in the Discussion section that systematically evaluates the five detection methods developed in this study. This table highlights their respective advantages, limitations, and potential applications. Please see lines 424-435.
C10: Lines 88–96: Please cite the appropriate references in this section.
R10: Done as suggested, please see lines 57, 60, and Reference section marked in blue.
C11: Unavailable web address: The address https://www.moa.gov.cn/hd/zqyj/index_1.htm is not accessible. If possible, please provide a screenshot of the content used from this site and include it in the article’s supplementary file.
R11: We have changed the websites into References. Please see line 96.
C12: Lines 111–114: Please include references for this section. If the transgenic rice mentioned was developed by the authors, more detailed information should be provided in the article’s supplementary file. Otherwise, supporting documents—such as publications related to the development of this resistant line and other relevant data—must be properly cited in the manuscript.
R12: Done as suggested, please see line 129 marked in blue.
C13: Line 112: Please include the catalog number of the kit used in this section. Also, mention the catalog numbers of all other kits used throughout the study.
R13: Done as suggested, please see lines 136, 146, 159, 172-173, 200 marked in blue.
C14: Please provide references for all protocols mentioned in the Materials and Methods section, especially the original articles or manuals from which the protocols were derived.
R14: Done as suggested, please see lines 129, 157, 188, 207 marked in blue.
C15: Figure 2: Clearly indicate the control group in the figure. The article should include either positive or negative controls to properly interpret the experimental results.
R15: Done as suggested, please see lines 130-133, 277 marked in blue.
C16: Line 190: The format of the cited reference in this line does not match the format of other in-text citations. Please ensure that all references follow a consistent format and that a DOI (Digital Object Identifier) is included for each cited paper.
R16: Done as suggested, please see line 207 and all of the references in References section.
C17: Please include a general discussion in the Discussion section about the industrial applicability of this work and its results in monitoring rice fields globally.
R17: Done as suggested, please see the Discussion section, lines 436-447 marked in blue.
C18: In the Discussion section, compare the limitations of the methods used in this study.
R18: Done as suggested, please see the Discussion section, lines 448-453 marked in blue.
C19: The conclusion section is too long and includes repetitive content from other sections of the manuscript. Please rewrite this section to concisely summarize only the key experimental findings.
R19: Done as suggested, please see the Conclusion section, lines 454-465 marked in blue.
C20: The references used in the article are outdated. Please use more recent and up-to-date references to support the discussion and compare the obtained results with those reported in recently published articles from scientific databases.
R20: Done as suggested, please see the References section marked in blue.
Round 2
Reviewer 2 Report
Comments and Suggestions for Authors
Table 4 should be moved from the 'Discussion' section to the 'Results' section.
Author Response
C:Table 4 should be moved from the 'Discussion' section to the 'Results' section.
R:Done as suggested, please see Lines 328-340 marked in blue.
Reviewer 3 Report
Comments and Suggestions for Authors
Dear colleagues,
I have no further comments on this paper. It can be accepted for publication in this journal.
Best regards,
Rasouli. H
Author Response
C:I have no further comments on this paper. It can be accepted for publication in this journal.
R:Thank you for your review and decision. We appreciate your time and feedback.